# Mathematical Model Based on the Shape of Pulse Waves Measured at a Single Spot for the Non-Invasive Prediction of Blood Pressure

**Lukas Peter** [1],*, **Jan Kracik** [2], **Martin Cerny** [1], **Norbert Noury** [3] and **Stanislav Polzer** [4]

1   Department of Cybernetics and Biomedical Engineering, VSB-Technical University of Ostrava, 17.listopadu 2172/15, 708 00 Ostrava-Poruba, Czech Republic; martin.cerny@vsb.cz
2   Department of Applied Mathematics, VSB-Technical University of Ostrava, 17.listopadu 2172/15, 708 33 Ostrava-Poruba, Czech Republic; jan.kracik@vsb.cz
3   lab. INL, University of Lyon, UMR 5270 CNRS-INSA Lyon, 69 621 Villeurbanne, France; norbert.noury@insa-lyon.fr
4   Department of Applied Mechanics, VSB-Technical University of Ostrava, 17.listopadu 2172/15, 708 33 Ostrava-Poruba, Czech Republic; stanislav.polzer@vsb.cz
*   Correspondence: lukas.peter@vsb.cz

**Abstract:** Background: Continuous non-invasive blood pressure (BP) measurement is a desired virtue in clinical practice. Unfortunately, current systems do not allow one for continuous, reliable BP measurement for more than a few hours per day, and they often require a complicated set of sensors to provide the necessary biosignals. Therefore we investigated the possibility of proposing a computational model that would predict the BP from pulse waves recorded in a single spot. Methods: Two experimental circuits were created. One containing a simple plastic tube for model development and a second with a silicone molded patient-specific arterial tree model. The first model served for the measuring of pulse waves under various BP (70–270 mmHg) and heart rate (60–190 beats per minute) values. Four different computational models were used to estimate the BP values from the diastolic time. The most accurate model was further validated using data from the latter experimental circuit containing a molded patient-specific silicone arterial tree. The measured data were averaged over a window of one, three, and five cycles. Two models based on pulse arrival time (PAT) were also analyzed for comparison. Results: The most accurate model exhibits a correlation coefficient of r = 0.967. The Bland–Altman plot revealed standard deviations (SD) between the model predictions and measurement of 10, 8.3, and 7.5 mmHg for the systolic BP and 8.7, 7 and 6.3 mmHg for the diastolic BP (both pressures calculated for the averaging windows of one, three, and five cycles, respectively). The best of the used PAT based model exhibited a SD of 17, 16, and 15 mmHg for the systolic BP and 14, 13, and 12 mmHg for the diastolic BP for the same averaging windows. Discussion: The proposed model showed its capability to predict BP accurately from the shape of the pulse wave measured at a single spot. Its SD was about 50% lower compared to the PAT based models which met the requirements of the Association for the Advancement of Medical Instrumentation.

**Keywords:** continuous non-invasive blood pressure; experimental model; pulse wave analysis; blood pressure prediction

---

## 1. Introduction

Continuous non-invasive blood pressure (BP) assessment is a desired virtue in clinical practice to monitor patients. For clinicians the BP is important parameter describing patient state in the case of patients accepted to hospitals [1,2]. Further, there are also strong arguments for BP home monitoring to

prevent various diseases. It has been shown that hypertension is a strong and independent risk factor of coronary heart disease [3], heart failure [4], peripheral arterial disease [5] or stroke disease [3,6,7]. This explains why most medical examinations include BP measurement.

Moreover, a more specifically defined hypertension (i.e., evening, morning episodic) was linked to several diseases recently [6–9]. To estimate it, however, more frequent and preferentially self BP measurement is necessary. However, BP is monitored usually at discrete time intervals by using an inflatable cuff placed around the arm, which may be uncomfortable for the patient in long term monitoring and represents a source of many mistakes due to inappropriate cuff size and/or BP variability between arms [10]. As an alternative measurement method, the volume clamp method based on plethysmography measurement [1,2] has been proposed. This method is clinically used but shows some disadvantages and limitations. Measurement is not comfortable for the patient and its accuracy decreases in time [11], so it is not applicable for more than several hours.

A possible solution of the continuous non-invasive BP assessment may be based on relation between BP and other patient/specific signals measured non-invasively, such as the electrical activity of the heart [12] and the pressure wave which propagates along the arterial system due to arterial elasticity. Firstly, the researchers tried to estimate the BP exploiting the relation between the electrical activity of the heart and the time in which the pulse wave reaches a certain point in the arterial system (pulse transit time (PTT)) [13]. This relation was, however, rather weak [11,14,15], so it was further refined by respecting the pre-ejection period (PEP) of the heart cycle. More recent algorithms operate mainly with the so-called pulse arrival time (PAT), which is the delay between the R-point on the ECG curve and the time the pulse wave reaches certain place [11,14–17]. These models show a very good correlation between BP and PAT on a test group of people but, on the other hand, they are highly dependent on calibration [15] and show often too large variability [11,15,18,19] for a clinical application. There are several reasons for that.

The arterial homoeostasis is balanced by several mechanisms such as vasodilatation and vasoconstriction for short term (order of seconds to minutes) and growth and remodelling processes in the long term (order of months to years). Further, it is affected externally, by muscle contractions and body positions. On the contrary, the current algorithms are mostly based on some variation of the Moens–Korteweg equation [18,20] which describes the pulse wave velocity (PWV) in a straight tube with constant diameter made of homogeneous linearly elastic material. However, it is a very rough approximation of a real arterial system consisting of hyperviscoelastic curved arteries with highly variable diameter and variable systemic vascular resistance [21]. This may be one reason for observed high standard deviations of these models [15] and in time decreasing accuracy. Moreover, all these algorithms require at least two sensors on the human body, for instance, to obtain PAT, plethysmography sensor and ECG measurement or two plethysmography sensors are used. These sensors needs to be perfectly synchronized during the whole measurements since the pulse wave velocity (PWV) is 4–16 m/s [21] and the measured time delay between the signals is in milliseconds [15]. Therefore any algorithm based on one sensor only would be much more robust and more suitable for everyday continuous BP measurement.

Therfore we focused on the possibility to estimate the BP from parameters that can be estimated from the shape of the pulse wave measured with a single sensor only. We employed our previously described experimental model [22] having its main advantage that many parameters can be adjusted as desired within a wide range, and the measurements can be performed as long as needed. Our goal was to propose an algorithm which would rely entirely on information extractable from a single pulse wave, thus avoiding the necessity of multiple sensors and their synchronization.

## 2. Methods

### 2.1. Experimental Model Setup

Since we are not aware of any precedent study estimating the BP purely from the shape of the pulse wave, we started our analysis with the simplest physical model where the artery is mimicked by astraight tube with 13 mm inner radius, and 1 mm wall thickness (see Figure 1A). It was made of silicon with Young's modulus of $E = 2.6$ MPa to ensure properties and pulse wave velocity comparable with human aorta, as well as applicability of the simple equations. Connection with the selected pump ensures correct operation of the physical model with water used as a substitute liquid for this part of the analysis.

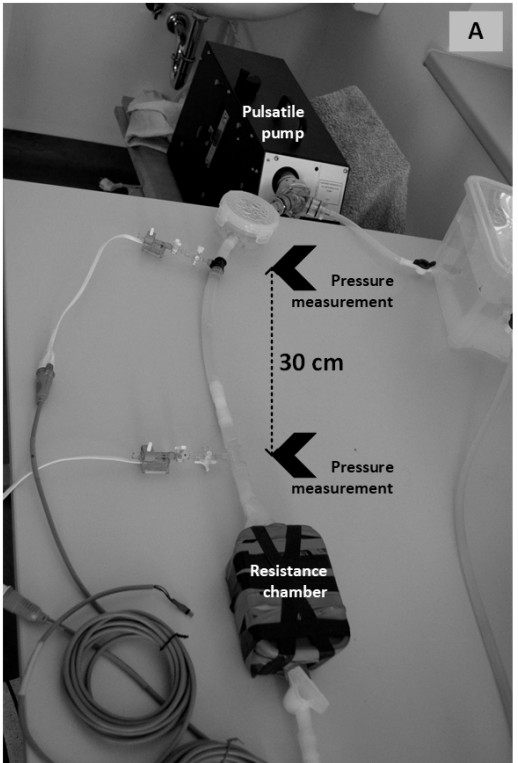
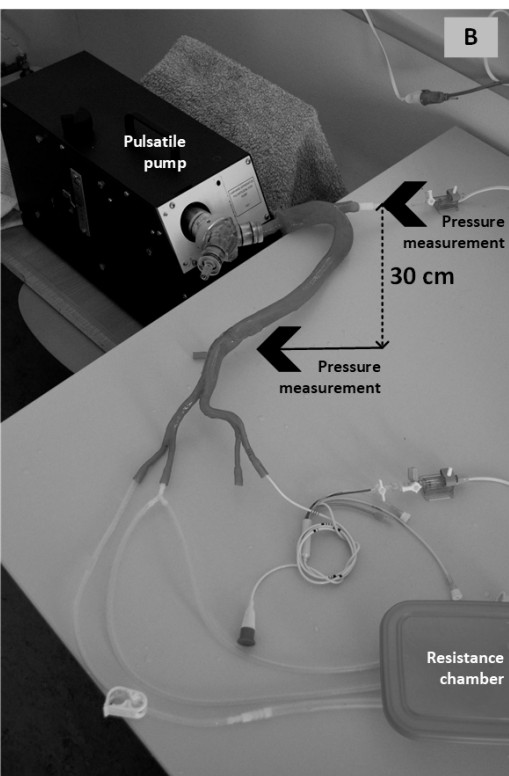

**Figure 1.** Experimental setting for development (**A**) and validation (**B**) of the computational model.

A pulsatile pump (Harvard Apparatus, Holliston, MA, USA) was used to mimic the heart. Note that this type of pump is clinically used as an external heart for large animals. The pump enables to adjust the basic hemodynamic parameters. Here we used the pulsatility of 25%/75% (time ratio of systole to diastole in one heart cycle) and the stroke volume of 15 ml. In our experiment, we varied the heart rate between 60 and 190 bpm which affected directly the blood pressure due to the constant systemic vascular resistance. The systemic vascular resistance was mimicked by a plastic box filled with glass balls. The glass balls inside reduce the pulsating blood flow to constant, so there is no change in the volume of the box. Thanks to this box, also a wave reflection was achieved [22]. It is noted the stroke volume is about five times lower compared to human heart so the systemic vascular resistance was adjusted by valves located prior to reservoirs accordingly to obtain the same blood pressure wave as in humans, see Figure 2. More details about the systemic vascular resistance setup can be found in our previous work [22].

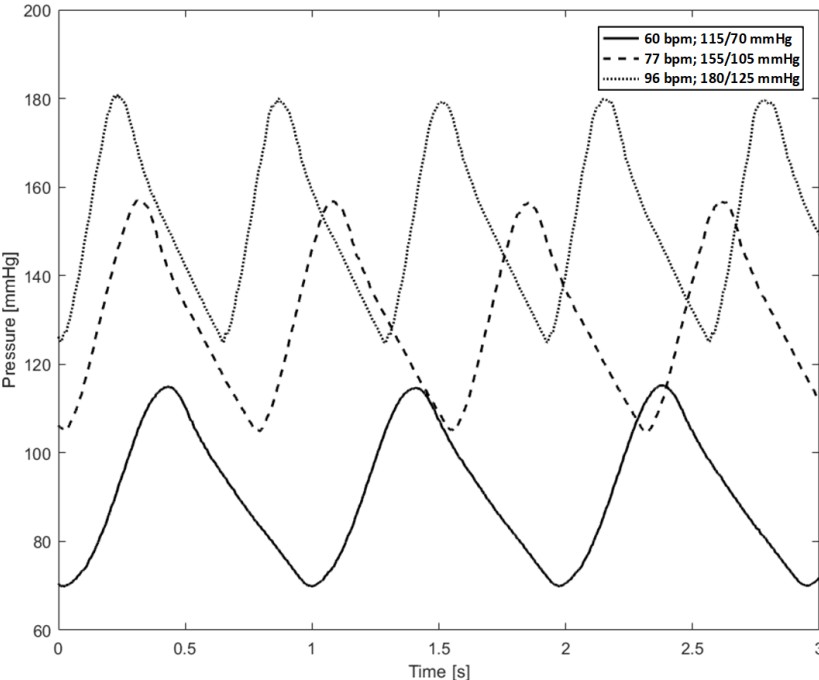

**Figure 2.** Pulse waves measured via an extravascular sensor for various heart rates and consequent blood pressures. It is showns the general shape of the pulse wave remains similar regardless of heart rate and Blood Pressure (BP).

Only pressure waves were measured because our experimental setup did not allow for diamater pulse wave measurements. Both waves, however, are known to have similar shapes at least qualitatively [23], especially in time domain as shown in Figure 3. Therefore it is possible to consider both waves fully commutable in the time domain and all further analyses are performed on pressure pulse waves, only.

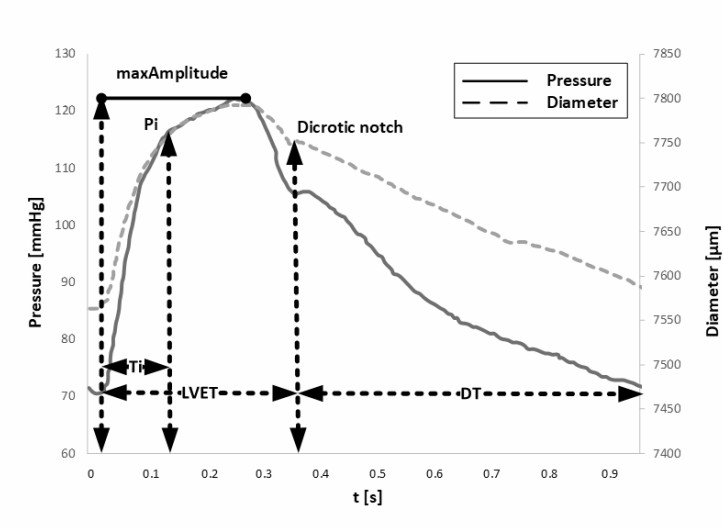

**Figure 3.** Comparison of aortic diameter and pressure pulse waves during one cardiac cycle (data from [23], page 162) shows similar shapes of both waves with qualitatively the same features in time domain (left ventricular ejection time (LVET), diastolic time (DT), travel time of the reflected wave (Ti)) were estimated.

The pressure signal was measured with two extravascular pressure sensors (DPT-6100, CODAN Medizinische Geräte GmbH & Co KG, Lensahn, Germany) connected to the tube via T-like connectors. One was placed at the beginning of the tube and the other at its end, in mutual distance 30 cm. Note that the second sensor served only to obtain data for assesment of the BP on the basis of PAT (see Section 2.3) for comparison and it was not used in our correlation analysis.

Measurement of the pulse wave was recorded at a sampling frequency of 1 kHz for 5 min during which the heart rate was gradually increased and decreased. Each heart rate frequency was set for few seconds from the minimum frequency to the maximum and back to minimum. Once the pulse waves were obtained, the time-related parameters left ventricular ejection time (LVET), diastolic time (DT) and travel time of the reflected wave (Ti) were estimated. The process is decribed in detail in our previous work [24] as well as in other works [25,26]. It is based on an analysis of signal derivatives as shown in Figure 4. Briefly, detection of foot and peak of the wave is based on the analysis of the first derivative of the measured signal. The foot of the signal is defined as the changes from negative to positive values of the first derivative of the signal, and vice versa for the peak. Pulse waves also include the dicrotic notch which occurs due to reflection of the wave from aortic valve when it is closing after systole and reflection of forward wave from bifurcation of artery. The dicrotic notch corresponds to the first maximum of second signal derivative after the peak of the signal. The inflection point Pi of encounter between the forward and backward waves is defined with the fourth derivative crossing zero after the first positive peak of the signal. The inflection corresponds to the beginning of the superimposition of backward wave (Pi). In young people, the inflection (Pi) occurs after the peak systolic pressure, while it moves before the peak systolic pressure with age [25].

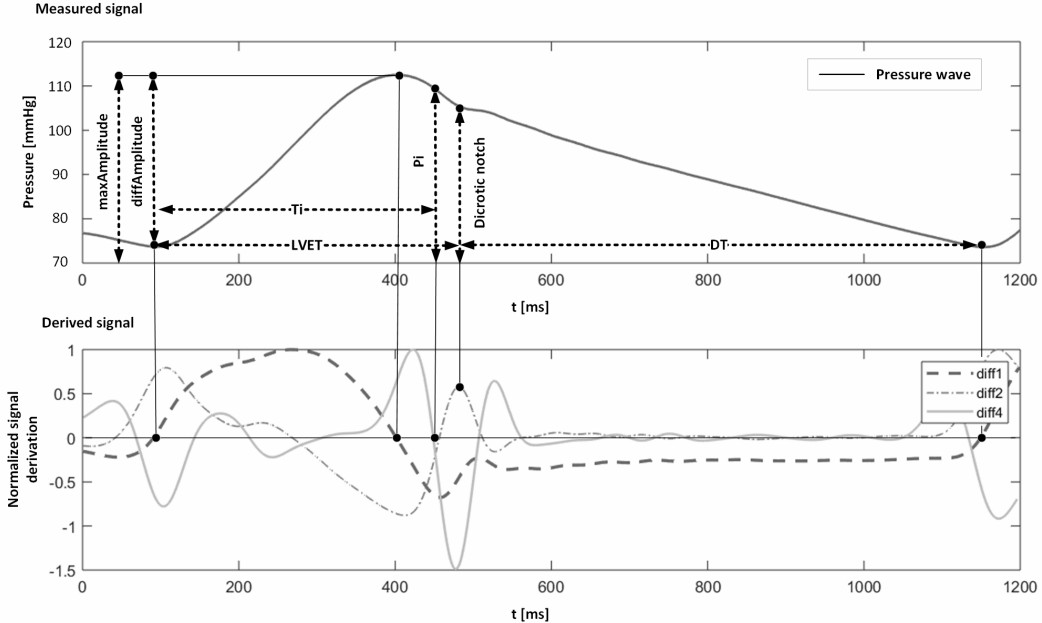

**Figure 4.** Pulse wave recorded on a patient-specific aorta (upper graph) and its first (diff1, Dashed line), second (diff2, Dash-dotted line) and fourth (diff4, Solid line) derivatives (lower graph) demonstrate how we measured the characteristic time parameters of the pulse wave (Ti, LVET, and DT) used for further analyses.

## 2.2. Correlation Analysis

The pressure waves recorded in a chosen point were analysed. Parameters described in Table 1 were extracted and correlation analysis (using Spearman's correlation) performed. Generally, we were searching for correlations between temporal pulse wave parameters DT, LVET and Ti (see Figure 4) and the extreme values (min and max) which represent diastolic and systolic BP respectively. Further,

mutual correlations between DT, LVET, and Ti were also estimated to identify only those parameters that are independently correlated with extreme values of the pressure wave. In total, correlations were evaluated in some 600 points, where each point represents one cardiac cycle.

**Table 1.** List of parameters measured experimentally.

| Parameter | [Unit] | Definition |
|---|---|---|
| minAmplitude | [V] | The lowest amplitude of the signal recorded in the pulse wave |
| Pi | [V] | The amplitude of the pulse wave corresponding to the point where the backward wave starts superimposing onto the forward wave |
| maxAmplitude | [V] | The maximum amplitude of the signal in systole |
| LVET | [ms] | Duration of the systolic phase, left ventricular ejection time |
| Ti | [ms] | Travel time of the reflected wave, the time delay of the backward waveform |
| DT | [ms] | Diastolic time, the time of the diastolic phase, from the dicrotic notch to the end of diastole |

Parameters showing strong correlations were further used to propose new equations on the basis of these results. These equations allow us estimation of the diastolic blood pressure, systolic blood pressure and mean arterial pressure (DBP, SBP and MAP) solely from the pulse pressure wave measured at the single spot. In this way four possible phenomenological models (see Table 2) were proposed, and the model with the highest correlation coefficient was used for further validation analysis.

**Table 2.** Various models and correlation coefficients of the fit to the measured DBP.

$$DBP = c_1 + c_2 \cdot DT \quad r[-] = -0.96 \tag{1}$$

$$DBP = \frac{1}{(c_1 + c_2 \cdot DT^2)} \quad r[-] = -0.95 \tag{2}$$

$$DBP = e^{(c_1 - c_2 \cdot \sqrt{DT})} \quad r[-] = -0.94 \tag{3}$$

$$DBP = c_1 + \frac{c_2}{DT} + c_3 \cdot DT^2 \quad r[-] = -0.97 \tag{4}$$

where $c_1$, $c_2$ and $c_3$ are calibration constants, which have to be obtained during through calibration procedure.

*2.3. Validation Analysis*

A straight tube used as the first model of the human arterial system is a very rough simplification applicable only to deliver correlations but it can hardly be used to test the description power of any proposed equation. Therefore the chosen equation was tested using data from the second type of the experimental model. This model was described before [22], thus only a brief summary is provided here.

The straight silicon tube was replaced with a molded model of arterial tree with patient-specific geometry (50 years old men without any obvious vascular diseases) containing whole aorta and other major elastic arteries as it was reconstructed from CT scannig. The lumen was 3D printed, and the wall was created by its repeatable diving into silicone solution to reach a thickness of 1 mm. The rest of the arterial tree was modeled with reservoirs mimicking its resistivity and compliance. The venous tree was modeled simply with another reservoir. The described geometry can be seen in Figure 1 right. Finally, it is noted we used solution of 60% water and 40% of glycerine (GLY) as blood analog here. This solution has viscosity of 4 mPa.s which matches the mean viscosity of blood in aorta.

Two sensors were used for measurement of the pressure wave in the aortic segment. A catheter was placed into the aortic segment (through femoral arteries, see Figure 1B) which was connected

directly to the pressure sensor described above. The second sensor was connected directly to the beginning of the printed part of the arterial tree (through carotid arteries, see Figure 1B).

Measurement was recorded for 5 min during which the heart rate was gradually increased from the 60 bpm by 10 bmp steps to 190 bpm. At each level, the data was recorded for 20 s after 5 s waiting for pressure wave stabilization. Consequently, the data from 630 pulse waves were analyzed.

The results obtained by the proposed equation were compared with these measured data considered as a ground truth set. DBP, SBP, and MAP were estimated using the best of four models presented in Table 2 from (i) each cardiac cycle as well as the average of (ii) three and (iii) five cycles. The motivation here is to decrease the effect of noise, and the averaging windows are chosen small enough to avoid smearing out significant differences in BP. The floating average is a standard tool for reducing noise in the processing of biosignals such as BP.

Finally, the accuracy of the obtained equation is compared with other known models published in literature, both linear and nonlinear ones. A linear equation was prposed by Baek et al. and by Choi et al. [17,27] as follows:

$$BP = c_1 + c_2 \cdot PAT \tag{5}$$

A nonlinear equation was proposed by Magder S. [28] as follows:

$$BP = c_1 + \frac{c_2}{PAT} \tag{6}$$

In both of these equations $c_1$ and $c_2$ are calibration constants.

Note that for our system, the PAT is identical with pulse transit time (PTT) because our pump does not have any pre-ejection period. Deviations between the measured and calculated SBP, DBP, and MAP were estimated using the Bland–Altman plot, and 95 % confidence intervals were estimated.

## 3. Results

Pulse waves recorded using straight tube setup for various BPs and heart rates are shown in Figure 2 while a typical pulse wave recorded using patient-specific aorta model is shown in Figure 4. All the curves have the same global shape which is a necessary condition, otherwise we could not expect the proposed model would work over a wide range of BPs and heart rates.

### 3.1. Correlation Analysis

Correlation plots are shown in Figure 5. Spearman's correlations between the investigated temporal and spatial variables are shown. Each cell with dots shows correlation between the given row variable and the correspoding column variable. Cells on the diagonal represent individual histograms of the row variables.

Strong correlations were found between all temporal parameters LVET, DT, Ti and both spatial SBP and DBP. However, the temporal parameters are also mutually correlated, so they all carry the same information, consequently, one parameter only is sufficient to estimate both DBP and SBP. For practical purposes, we have chosen the DT since the time of the diastole is the longest and thus, any measurement error has the least serious impact on accuracy.

Based on the results of correlation analysis, we plotted the data of BP vs. DT and fitted the data with several equations to find the model giving their best fit. The results are listed in Table 2. Model Equation (4) showa the highest $R^2$, and we show its capability to capture the DBP (and thus also the other BPs) is shown in Figure 6. Consequently, this model was chosen for validation analysis.

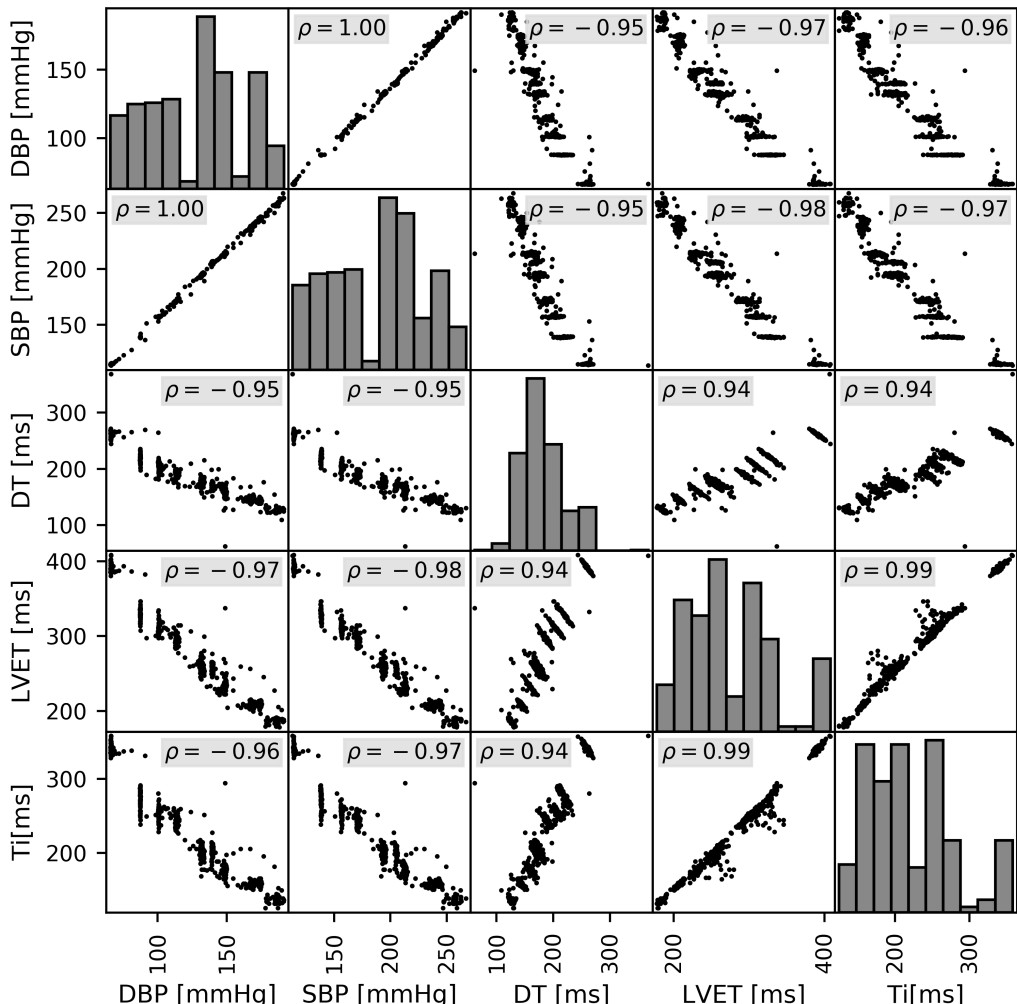

**Figure 5.** Correlations between the investigated temporal and spatial variables together with Rho values.

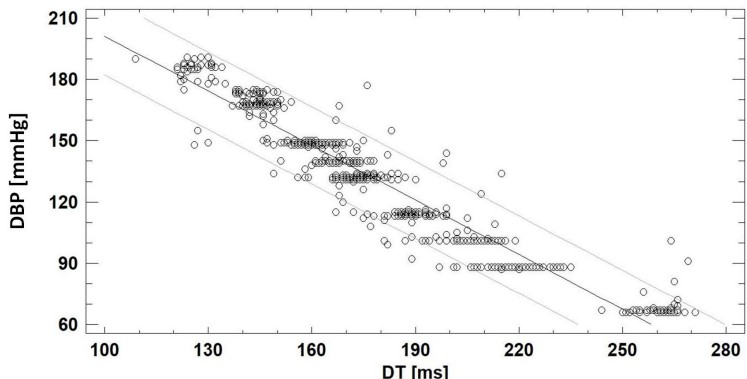

**Figure 6.** Capability of the model (Equation (4) to describe the measured DBP as a function of DT. The solid line indicates the mean value, while the dashed lines mark 95 % confidence interval.

*3.2. Validation Analysis*

Bland–Altman plots for comparisons between the proposed model Equation (4) and the experimentally measured values on the second (more realistic) experimental model are shown in Figure 7.

The analysis was made for non-averaged values as well as for the values averaged over moving windows of three (data not plotted) and five consecutive cycles (MA5). The standard deviation (SD) decreased with increasing size of the moving average window as shown in Figure 7. Specifically, SD was (10; 8.3; 7.5 mmHg) for SBP, (9.3; 7.3; 6.7 mmHg) for MAP and (8.7; 7; 6.3 mmHg)for DBP for one, three, and five cycles windows, respectively. On the contrary, use of PAT based models lead to significantly higher SDs. For the linear PAT model (Equation (5)) it was (19; 18; 17 mmHg) for SBP, (18; 16; 15 mmHg) for MAB and (17; 15; 14 mmHg) for DBP, respectively. A slightly better accuracy was obtained for the nonlinear PAT model (Equation (6)). Specifically, SDs were (17; 16; 15 mmHg)for SBP, (15; 14; 13 mmHg) for MAP and (14; 13; 12 mmHg) for DBP, respectively. Accuracy of the prediction of BP via (Equation (6)) is shown in detail in Figure 8 for not-averaged and averaged (over five cycles) data.

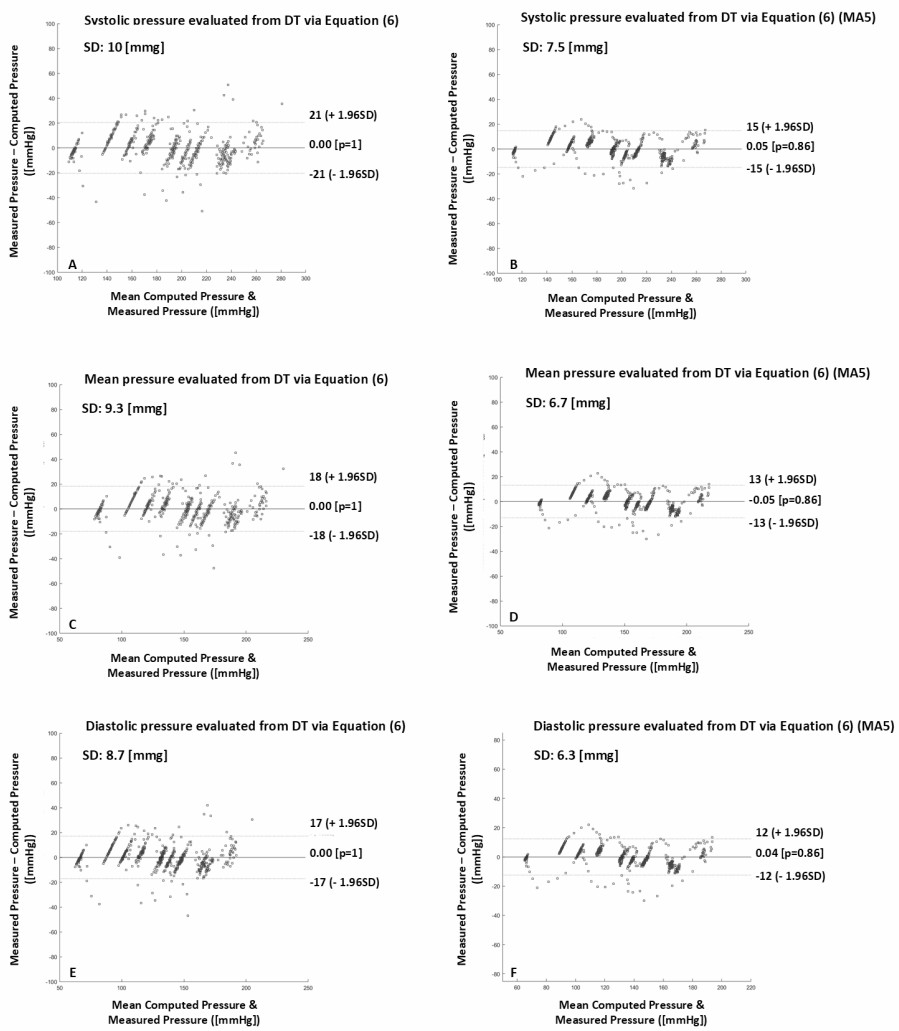

**Figure 7.** Bland Altman comparisons of BPs measured experimentally vs. those computed via the proposed model (Equation (4)). Comparison of SBP is shown (**A**,**B**), MAP (**C**,**D**) and DBP (**E**,**F**) for not-averaged values (**A**,**C**,**E**) and values averaged over five cycles (**B**,**D**,**F**). Averaging was made thanks to moving average method (MA5). In all the cases, there is no systematic deviation between the measured and computed BPs while standard deviation decreases when the response is averaged over more cycles.

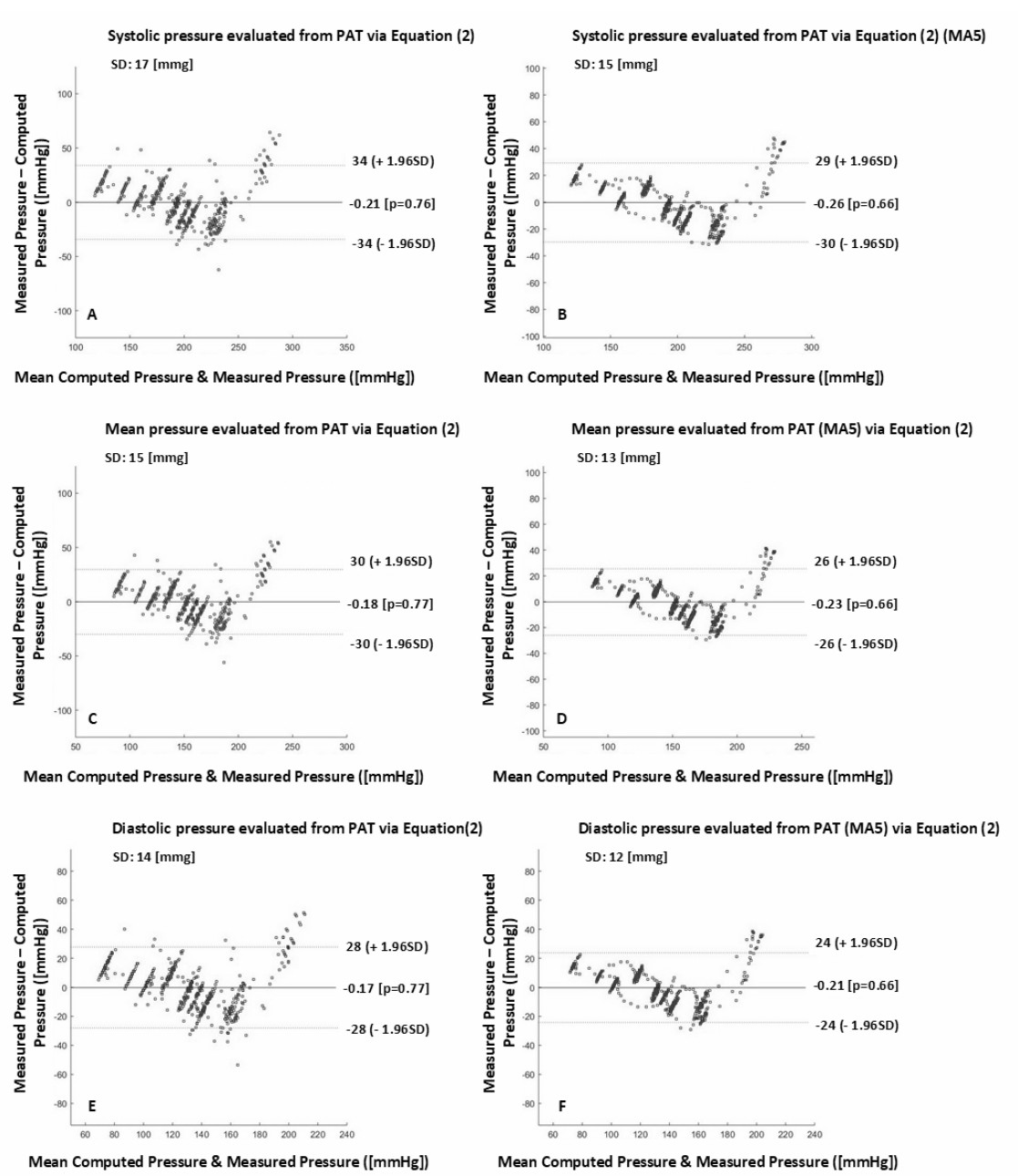

**Figure 8.** Bland Altman comparisons of BPs measured experimentally vs. those computed via model proposed by Magder S. [28] (Equation (6)). Comparisons of SBP is shown (**A**,**B**), MAP (**C**,**D**) and DBP (**E**,**F**) for non-averaged values (**A**,**C**,**E**) and averaged values over five cycles (**B**,**D**,**F**).Averaging was made thanks to moving average method (MA5). In all cases, there is no systematic deviation between measured and computed BPs, while the standard deviation decreases when the response is averaged over more cycles.

## 4. Discussion

In this study, we have proposed a new approach of how to estimate the BP without necessity of measuring the PAT (or PTT). Instead, we proposed a model (Equation (4)) which estimates the BP from the shape of the pulse wave, thus it does not require measurement of pulse wave propagation. Consequently, it requires a single pressure sensor only which reduces the number of sources of potential errors due to wrong sensor attachments or synchronization mismatch. Further, thanks to the

experimental setup we were able to investigate pulse wave shapes across a wide range of BPs which is naturally not possible in humans, due to ethical reasons.

Finally, comparison between in vivo recorded pulse waves (see Figure 3) and pulse waves obtained with our experimental setup (see Figure 2 for the straight tube setup and Figure 4 for a patient-specific geometry of arterial tree) shows they match qualitatively in time domain except for the position of Pi which is not used in our analysis anyway. Although the dicrotic notch was rather suppressed in the straight tube setup (Figure 2), we were still able to detect it thanks to derivatives of signals, and it was nicely pronounced in the patient-specific aorta setup (see Figure 4). This confirms the proper setup of our experimental loop.

Analysis of the pulse wave revealed that all the shape-based parameters depend linearly on each other (see Figure 5). This means the pulse wave shape remains similar across a wide range of BP which confirms our initial assumption. Theoretically it also means the BP can be predicted with the same accuracy from any of these parameters. In practice, this conclusion should be interpreted with caution since it was not validated with differential geometry which should be certainly done in future studies.

The proposed phenomenological modell was validated on a completely different geometry, and the results show its predictive capability is still very good. It was capable to predict BP with $SD < 10$ mmHg. These results were further improved by using a floating average. With that, we were able to further decrease the variability of BP prediction by up to 25 % to $SD < 8$ mmHg when averaged across five cycles. This is a very important result because it meets requirements given by the Association for the Advancement of Medical Instrumentation (AAMI) (85% of measurements with SD 8 mmHg). The more strict accuracy requirements of British Hypertension Society (BHS) (75% of measurement within 10 mmHg) were unfortunately slightly missed, so a larger averaging window would be necessary.

On the other hand, the averaging window should not be too large. Otherwise, we are losing information about BP changes with the range of the window size. This gives us an upper limit for which the averaging is applicable (11–15 cycles).

A comparison of our model with the PAT based models Equations (5) and (6), is also favorable, it shows the proposed model exhibit about half SD for all the analyzed BPs and averaging windows (cp. Figures 7 and 8). There is a large variability in reported errors of the PAT based models. Comparison of our results to available literature shows there is large variability from SD as low as 4 mmHg [17] up to 22 mmHg [15]. Recent review independently compared available PAT based models and reported SD varying between 11 to 14 mmHg for SBP to 19 to 22 mmHg for DBP [15]. Our results obtained from PAT based models shows slightly higher SD for SBP ($\approx$15 mmHg) and significantly lower SD for DBP ($\approx$11 mmHg) see Section 3.2 and Figure 8. Better performance in case of DBP may be caused by constant systemic vascular resistance used in our experimental model. Nevertheless none of these results comply with requirements of BHS and AAMI. But even if PAT based models would have comparable accuracy as proposed model, our model would still be more robust due to the only one required sensor compared to complicated set of sensors necessary for the most accurate BP measurement via PAT based models [17].

Despite the promising results, it is important to note further steps are necessary to fully validate the model. First, the predictive capability should be tested on a more advanced experimental model. For instance, variable systemic vascular resistance (SVR), non-constant pulse pressure or more patient-specific geometries of the aortic tree can be used to check the accuracy in more realistic conditions and to assess the impact of inter-patient variability. This step will anticipate a final validation on a real patient, where it will be analyzed if and how frequently it is necessary to adjust the proposed model to maintain the desired accuracy.

Our results also do not support the hypothesis the inaccuracy of the PAT based model is due to motion artifacts or sensor movements [15,29] since our model does not suffer from them. Instead we speculate the source of this variability may be in the complicated geometry and/or viscoelastic nonlinear behavior of the material (which remains similar for our model compared to a real patient)

where the actual diameter is neither linearly dependent on the BP, nor it is a simple consequence of actual BP, but it depends also on previous states of the system. On the contrary, any model based purely on some modification of Moens–Korteweg equation neglects these phenomena and fails when BP is unstable. However, more research is needed to analyze this problem.

Our results should be viewed with respect to their limitations. In our case, it is, of course, the experimental model which does not have some of the relevant features observed in human body. One of them is a nonlinear anisotropic viscoelastic non-homogeneous material of the arterial walls the stiffness of which may change due to smooth muscle cell contraction. These complex properties were replaced in our experimental model with a passive isotropic linear homogeneous material, which affects certainly the pulse wave shape, especially under non-constant conditions (during exercise or heat/cold exposure, etc.). The second limitation is our pump, which has the stroke volume of 15 ml only, i.e., about five times lower compared to humans. This limitation was compensated by adjusting the systemic vascular resistance to obtain the same pulse wave as in humans which however prevented variation of systemic vascular resistance during measurements. Pulsatile pump with higher stroke volume could overcome this limitation. The third limitation of the study is in having only one patient-specific geometry of the arterial tree for validation. More geometries would be certainly helpful. On the other hand, the fact we have developed our model for BP estimation on results from a straight tube, and yet the accuracy of predicting BP on completely different geometry and material support our assumption the model is relatively robust and can be used for further validation on real patients. The fourth limitation lies in pressure wave measurement and model evaluation for non-invasive pulse wave measurements. We evaluated the time properties of the wave on the basis of similarity between pressure and pulse waves (pulse waves can be measured noninvasively by photoplethysmography sensor). In the future, we expect to be able to evaluate the mathematical model for noninvasive pulse wave measurements, see Figure 3.

## 5. Conclusions

We have proposed a novel mathematical approach for cuffless BP monitoring. This approach is based on the analysis of time-related features of the pulse wave. The mathematical model obtained from measurements on a physical model with a straight plastic tube was validated via patient-specific, silicone molded model of arterial tree. Validation proved the proposed model achieved about double accuracy compared to the traditional approach based on PAT. The accuracy depended on the size of the moving average window and the proposed approach met the requirements given by the Association for the Advancement of Medical Instrumentation when averaged over five cycles.

The proposed model is also much more robust compared to models based on PAT because it relies on measurement at a single spot and does not require evaluation of a very short time delay of signals from two spots.

Consequently, the proposed model is promising and should be further validated on both more realistic experimental models and real patients.

**Author Contributions:** L.P. designed and performed the experiments. J.K. did the statistical analysis and S.P. wrote and edited the manuscript with input and editing from M.C. and N.N. All authors have read and agreed to the published version of the manuscript.

**Funding:** This work was supported by Czech Science Foundation grant Nr. 19-22426Y and by the project Biomedical Engineering Systems XV Nr. SV450994'.

**Acknowledgments:** The authors would like to thank the editors and the reviewers for their constructive comments and suggestions.

**Conflicts of Interest:** The authors declare no conflict of interest.

## Abbreviations

The following abbreviations are used in this manuscript:

BP     Blood Pressure
DBP    Diastolic Blood Pressure
SBP    Systolic Blood Pressure
MAP   Mean Arterial Pressure
DT     Diastolic Time
LVET   left ventricular ejection time
PTT    Pulse Transit Time
PAT   Pulse Arrival Time

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
