# Peer review of "Mathematical Model Based on the Shape of Pulse Waves Measured at a Single Spot for the Non-Invasive Prediction of Blood Pressure"

_processes, doi:10.3390/pr8040442_

Round 1
Reviewer 1 Report
- Title, Non-invasive prediction of blood pressure based on shape of pulse wave measured at single spot -> Evaluation of mathematical approaches based on … for …
- Keywords: CNIBP -> full spell; analysis -> too broad, deletion better.
- P1L23, Continuous non-invasive Continuous non-invasive -> Continuous non-invasive
- Figure 1, not clear enough. Please use illustrations instead of photos.
- P5L99, at a frequency of 1 kHz -> at a sampling frequency of 1 kHz
- P5L100, “was gradually increased” only? No gradually decreased? Bidirectional manipulation may be better.
- Experiment protocol should be given.
- Figure 4, its first (Dashed line) -> its first (diff1, Dashed line), and others
- Table 1, Abbreviation [unit] -> Unit
- Table 1, definition -> Definition
- Table 2, various models and correlation coefficient …-> Various models and correlation coefficients …
- Table 2, why DBP only? No results for SBP?
- Table 2, move to the position before Section 2.3
- Figures 6, why DBP only? No results for SBP?
- Figures 7, 8, what is (MA5)?
- P12L226, proposed models exhibits -> proposed models exhibit
- It seems more realistic experimental model should be also setup and examined.
- Real data for validation of the proposed method should be collected from patients.
Author Response
We would like to thank the Reviewer for reading our manuscript carefully again. Mentioned comments were found valuable and further improved our manuscript. Point to point response is attached below:
Comments and Suggestions for Authors
Title, Non-invasive prediction of blood pressure based on shape of pulse wave measured at single spot -> Evaluation of mathematical approaches based on … for …
The suggestion was partly accepted. The changed title is:
“Mathematical model based on shape of pulse wave measured at single spot for non-invasive prediction of blood pressure”
Keywords: CNIBP -> full spell; analysis -> too broad, deletion better.
Keywords were changed as suggested
P1L23, Continuous non-invasive Continuous non-invasive -> Continuous non-invasive
Corrected.
Figure 1, not clear enough. Please use illustrations instead of photos.
Figure 1 was found sufficient by the other 3 reviewers. However, we added at least the reference to our previous work where the scheme of this loop is presented.
P5L99, at a frequency of 1 kHz -> at a sampling frequency of 1 kHz
Corrected.
P5L100, “was gradually increased” only? No gradually decreased? Bidirectional
manipulation may be better. Experiment protocol should be given
The reviewer is right and we did it in such away. It was only not included in the text by mistake. It is corrected now. The experimental procedure, however, is the same as for the validation analysis as described starting at line L147 is described. The manuscript now states that more clearly.
Figure 4, its first (Dashed line) -> its first (diff1, Dashed line), and others
Corrected.
Table 1, Abbreviation [unit] -> Unit
Corrected
Table 1, definition -> Definition
Corrected.
Table 2, various models and correlation coefficient …-> Various models and correlation coefficients …
Corrected.
Table 2, why DBP only? No results for SBP?
A Typo. Of course that the correlation was evaluated for all pressures. Corrected
Table 2, move to the position before Section 2.3
Moved as suggested.
Figures 6, why DBP only? No results for SBP?
We do have results for SBP as well but they look practically identical so we do not see any added value of plotting them.
Figures 7, 8, what is (MA5)?
MA means moving average. The abbreviation is now introduced in 3.2 Validation analysis.
P12L226, proposed models exhibits -> proposed models exhibit
Corrected.
It seems more realistic experimental model should be also setup and examined.
We fully agree and that is why further model development is discussed in the discussion section starting at line L234.
Real data for validation of the proposed method should be collected from patients.
We fully agree. Validation on clinical data is necessary next step as we mention at line 238 and such analysis is now in progress. However even if we had this data already we could hardly squeeze that within a single paper. It took us about 4000 words (without references) only to describe methodology and experiments performed in-silico (current paper). The description of the clinical study we are working on seems to add another 4000 words. Therefore we do not plan to add this to our current paper
Reviewer 2 Report
The problem is very well selected. A simple, and accurate, method for blood pressure prediction is urgently needed to be introduced in multiple fields to help monitoring, and even improve, treatments. In summary, the proposed method allows to estimate this metric from parameters that can be assessed from the shape of the pulse wave, measured by only one sensor. This is the novelty of the work that presented a mathematical equation for blood pressure prediction. While the study is very interesting, some remarks remain.
Remarks:
-The description of the methods in the abstract is confusing. It is indicated that “Two experimental models were created.”. The next sentence starts with “One containing a simple plastic…”, however, the “second” is not specified, creating a confusion.
-Some sentences are written in a non-standard way, as an example: “Better of used PAT based model exhibited…”, or have grammatical errors, as an example: “Continuous non-invasive Continuous non-invasive blood pressure (BP) estimation…”. English proof checking should be performed.
-Sentences such as “Besides obvious reasons in the case of patients…” are not suitable for a scientific work. A formal justification should be presented and not based on “obvious”.
-Sentences, such as “…pulse wave velocity (PWV) in the straight tube of the constant diameter made of homogeneous linearly elastic material which is only very rough approximation of the real arterial system resulting in high standard deviations and in time decreasing accuracy as described above.”, require a reference to corroborate the claim.
-A formal performance comparison with the results reported by other works, based on different methods, should be introduced in the discussion.
-It is indicated that plethysmography measurement “accuracy decreases in time [11], so it is not applicable for more than several hours.”. Is the proposed model suitable for multiple hour analysis? If the answer is yes, then this is a key element and should be demonstrated. If the answer is no, then the novelty of the work is diminished.
Author Response
We would like to thank Reviewer for reading our manuscript carefully again. Mentioned comments were found valuable and further improved our manuscript. Point to point response is attached below:
Comments and Suggestions for Authors
The problem is very well selected. A simple, and accurate, method for blood pressure prediction is urgently needed to be introduced in multiple fields to help monitoring, and even improve, treatments. In summary, the proposed method allows to estimate this metric from parameters that can be assessed from the shape of the pulse wave, measured by only one sensor. This is the novelty of the work that presented a mathematical equation for blood pressure prediction. While the study is very interesting, some remarks remain.
We thank the Reviewer for the supportive comment
Remarks:
-The description of the methods in the abstract is confusing. It is indicated that “Two experimental models were created.”. The next sentence starts with “One containing a simple plastic…”, however, the “second” is not specified, creating a confusion.
The second model description was added to the abstract.
-Some sentences are written in a non-standard way, as an example: “Better of used PAT based model exhibited…”, or have grammatical errors, as an example: “Continuous non-invasive Continuous non-invasive blood pressure (BP) estimation…”. English proof checking should be performed.
We are sorry for those errors. The revised manuscript was proofread.
-Sentences such as “Besides obvious reasons in the case of patients…” are not suitable for a scientific work. A formal justification should be presented and not based on “obvious”.
Sentence was reformulated.
-Sentences, such as “…pulse wave velocity (PWV) in the straight tube of the constant diameter made of homogeneous linearly elastic material which is only very rough approximation of the real arterial system resulting in high standard deviations and in time decreasing accuracy as described above.”, require a reference to corroborate the claim.
The reviewer is right that this formulation was too strong. The sentence was reformulated and reference added.
-A formal performance comparison with the results reported by other works, based on different methods, should be introduced in the discussion.
The mentioned part for the discussion was extended as Requested.
-It is indicated that plethysmography measurement “accuracy decreases in time [11], so it is not applicable for more than several hours.”. Is the proposed model suitable for multiple hour analysis? If the answer is yes, then this is a key element and should be demonstrated. If the answer is no, then the novelty of the work is diminished.
Here the honest answer is “we do not know yet”. The reviewer is perfectly right this ability would be a great advantage. However, we are not that far yet and we fully acknowledge more work needs to be done to validate our method. In this study, we proposed a method that does not require two sensors and is more accurate than other methods on the experimental model. We do not claim we can do better also in long term anywhere in the manuscript. We are aware that this needs to be tested first. We believe our model has a good chance to perform better than PAT based models but it is the only speculation without experiments so we do not comment on that in the manuscript.